# Does amiodarone impact on apixaban levels? The effect of amiodarone on apixaban level among Thai patients with non-valvular Atrial Fibrillation

**Sutee Limcharoen**[1], **Sarawuth Limprasert**[2], **Pornwalai Boonmuang**[3]*,
**Manat Pongchaidecha**[3], **Juthathip Suphanklang**[3], **Weerayuth Saelim**[3],
**Wichai Santimaleeworagun**[3], **Piyarat Pimsi**[3]

1 Department of Pharmacy, Faculty of Medicine, Vajira Hospital, Bangkok, Thailand, 2 Division of Cardiology, Department of Medicine, Phramongkutklao Hospital, Bangkok, Thailand, 3 Department of Pharmaceutical Care, Faculty of Pharmacy, Silpakorn University, Nakhon Pathom, Thailand

* boonmuang_p@su.ac.th

## Abstract

**Data Availability Statement:** All relevant data are within the paper.

### Background

Apixaban and amiodarone are drugs used for non-valvular atrial fibrillation (NVAF) in routine practice. The evidence about apixaban plasma levels in patients who receive apixaban with amiodarone, including bleeding outcomes, has been limited. This study aimed to compare the apixaban plasma levels and bleeding outcomes between apixaban monotherapy and apixaban with amiodarone groups.

### Methods

This study was a prospective, observational, and single-center research which was conducted from January 2021 to January 2022 in NVAF patients who received apixaban at a tertiary care hospital located in the center of Bangkok, Thailand.

### Results

Thirty-three patients were measured for their median (5th–95th percentile) apixaban plasma levels. The trough of apixaban plasma level ($C_{trough}$) were 108.49 [78.10–171.52] and 162.05 [87.94–292.88] µg/L in the apixaban monotherapy and apixaban with amiodarone groups, respectively (p = 0.028). Additionally, the peaks of apixaban plasma level ($C_{peak}$) were 175.36 [122.94–332.34] and 191 [116.88–488.21] µg/L in the apixaban monotherapy and apixaban with amiodarone groups, respectively (p = 0.375). There was bleeding that occurred in 7 patients (21.21%); 5 patients in the apixaban monotherapy group and 2 patients in the apixaban with amiodarone group, respectively.

**Funding:** This MS received fund from Faculty of Pharmacy, Silpakorn University (Reference No. 003/2564). The funders had no role in study design, data collection and analysis, decision to publish, or preparation of the manuscript.

**Competing interests:** The authors have declared that no competing interests exist.

## Conclusions

Amiodarone may increase the peaks and troughs of apixaban plasma levels. The co-administration of apixaban with amiodarone is generally well tolerated. However, the careful observation of bleeding symptoms in individual cases is necessary to ensure safety.

## Introduction

Atrial fibrillation (AF) is supraventricular tachyarrhythmia which causes an important health problem; the AF prevalence ranges from 0.4% to 2.2% and increases with age in Thailand [1]. AF is related to cardioembolic ischemic stroke [2]. Ischemic stroke is one of the most serious complications and the leading cause of death and disabilities in AF patients [3]. The three major goals of treatment are rate control, rhythm control, and thromboembolism prevention [4]. Thus, oral anticoagulants, including vitamin K antagonist (warfarin) and non-vitamin K antagonist (e.g., dabigatran, rivaroxaban, apixaban, and edoxaban), have been prescribed to prevent stroke in AF patients [5]. Apixaban is a direct factor of Xa inhibitors; landmark studies have shown its greater benefits compared with warfarin and decreased intracranial hemorrhage [6, 7]. In addition, apixaban is a rapid onset-offset drug, with convenient dosing, with no need for international normalized ratio (INR) monitoring [8]. However, the issue of drug–drug interaction poses concern because apixaban is a P-glycoprotein (P-gp) substrate and metabolized by cytochrome P450 (CYP) 3A4 [9]. Amiodarone is a class III antiarrhythmic agent classified by Vaughan Williams; it is used widely to treat ventricular arrhythmia and supraventricular arrhythmia including AF [10, 11]. This medication has been used for rate control and cardioversion among AF patients [4]. Amiodarone also interacts with a large variety of medications, and many of these drug–drug interactions result from the inhibition of CYP1A2, CYP2C9, CYP2D6, CYP3A4, and P-gp [11, 12]. The trend of apixaban use in Thailand has increased, due to the limited supply of warfarin. Thus, the combination of amiodarone and apixaban can occur in routine practice. Theoretically, the pharmacokinetic interaction between apixaban and amiodarone may result in increased apixaban plasma levels and bleeding [13]. Clinical practice guidelines reported the effect of amiodarone on the increase in area under the curve (AUC) of dabigatran, edoxaban and rivaroxaban (12%–60%, 40% and minor effect, respectively), but limited pharmacokinetic data are available for the apixaban group [14]. Therefore, the interaction between amiodarone and apixaban has been reported in caution. The data finding from the previous observational studies is controversial, some studies reported that clinical outcomes of amiodarone combined with apixaban did not increase the risk of stroke and major bleeding compared with warfarin, while other studies reported the apixaban levels only [15–18]. However, two large observational studies showed an increased risk of bleeding when using amiodarone with apixaban [19, 20]. Certain limitations have raised the concern about these drug–drug interactions in Thailand in various ways. Differences in patient demographics, such as body size, comorbidities, and co-medication, may influence the safety of apixaban profile [8]. Therefore, the benefit-risk profile of apixaban combination with amiodarone agents in real-world situations may differ from clinical practice guidelines [14]. Currently, the real-world evidence of drug–drug interactions between apixaban and amiodarone and their effect on apixaban level, anti-factor Xa (anti-FXa) activity, and bleeding outcomes is limited. This study aimed to compare the effect of amiodarone on the apixaban plasma levels and bleeding outcomes among non-valvular AF (NVAF) patients in routine care at a tertiary hospital in Thailand.

## Materials and methods

### Study design and setting

This study was a prospective, observational, and single-center research which was conducted from October 2020 to September 2022 among patients receiving apixaban at Phramongkutklao Hospital, a tertiary care hospital located in the center of Bangkok, Thailand. The study protocol was approved by the institutional review board of the Royal Thai Army Medical Department and Phramongkutklao Hospital (Approval No. S054h/63) and registered in Thai Clinical Trials Registry. The identification number is TCTR 20201005003 and this study was retrospectively registered.

### Study participants

Patients were eligible for recruitment if they were $\geq$ 18 years old, had received apixaban for at least 7 days, diagnosed with NVAF, and provided written informed consent. Patients were excluded if they denied providing informed consents, had moderate to severe mitral valve stenosis, mechanical valve replacement, Child–Pugh class C, chronic liver disease, or end-stage renal disease with or without dialysis or kidney transplantation, pregnant or breastfeeding, unable to perform self-care without a caregiver, or incompatible to receive apixaban and concomitantly using strong CYP3A4 and P-gp inhibitors. All patients who were included, were already receiving apixaban with or without amiodarone before enrollment based on the actual use of amiodarone before inclusion. The adherence to the use of these medications within 7 days before enrollment were determined by telephone and pill count method. After that patients were categorized into two groups as follows: apixaban monotherapy (control group) and apixaban with amiodarone groups. In addition, all patients received an appropriate apixaban dose based on clinical practice guidelines: the standard dose of apixaban is 5 mg twice daily. When patients have at least 2 of the following characteristics: age $\geq$ 80 years, body weight $\leq$ 60 kg, or serum creatinine (SCr) $\geq$ 1.5 mg/dL the recommended dose of apixaban is 2.5 twice daily [14].

### Data collection

Demographic data including age, gender, body weight, SCr, creatinine clearance (CrCl) by Cockcroft-Gault equation, comorbidities, relevant information about their AF (CHA$_2$DS$_2$-VASc, and HAS-BLED score) were extracted from the hospital database. In addition, specific information about apixaban and amiodarone therapy including doses and duration of usage were obtained.

Two blood samples were collected for apixaban level measures at steady state; 1) at recruitment taken before taking their morning dose of apixaban in the morning, 2) the peak level was taken after taking apixaban within 2–4 hours. After that, blood samples were collected into two 3.2% citrated tubes to measure both the trough and peak concentrations. The blood samples were centrifuged immediately for 15 mins at 2500–3000×g [14–16, 21]. Apixaban plasma levels were measured by chromogenic assay, which were performed with the BIOPHEN$^{TM}$ heparin LRT kit (Hyphen BioMed, Neuville-sur-Oise, France) and analyzed by a Sysmex CS 2500 System (Siemens Health Care, Milan, Italy). This assay was calibrated with commercial apixaban (sensitivity range 0–600 ng/mL) and calibrated with LMWH (Low Molecular Weight Heparin) sensitivity range about 0–1.75 international units per milliliter (IU/mL). Apixaban levels were compared to the expected ranges of 69–321 ng/mL and 34–230 ng/mL for the peak and trough levels, respectively [14]. The anti-Xa levels were measured by the same method as

the apixaban levels, using indirect method to standardize in the LMWH scale. All reagents and instruments were used in accordance with the manufacturers' instructions.

## Outcomes of interest

The primary outcomes compared the trough and peak of apixaban plasma levels and anti-FXa activity at the steady state between the control and apixaban with amiodarone group. Secondary outcome was occurrence of bleeding based on the International Society on Thrombosis and Haemostasis (ISTH) definition, as follows: major bleeding was defined as follows: (1) fatal bleeding; and/or (2) symptomatic bleeding in a critical area or organ (intracranial, intraspinal, intraocular, retroperitoneal, intra-articular or pericardial, or intramuscular with compartment syndrome); and/or (3) bleeding causing a fall in hemoglobin level of 20 g/L or more, or leading to transfusion of two or more units of whole blood or red cells; clinically relevant non-major bleeding (CRNMB) was defined as follows: any overt bleeding requiring a medical intervention (hospitalization, surgery or interventional procedure, further diagnostic imaging, laboratory test, or specialist evaluation) and/or treatment discontinuation, and not meeting any of the criteria for major bleeding [22]. After the blood samples were reported manual electronic chart reviews were performed and all patients were followed until the first occurrence of bleeding outcome or stopped using apixaban or until **1** year.

## Statistical analysis

Statistical analysis was performed using the Statistical Package for the Social Sciences (SPSS) statistics version 27.0. (IBM Corp, Armonk, NY). All variables were analyzed using descriptive statistics to determine the frequencies with percentages for categorical variables, while continuous variables were expressed in mean ± standard deviations (SDs) or median with interquartile range (IQR). The apixaban plasma levels were reported in median [$5^{th}$– $95^{th}$] percentiles based on the **2021** European Heart Rhythm Association Practical Guide on the Use of Non-Vitamin K Antagonist Oral Anticoagulants in Patients with Atrial Fibrillation [14]. All patients were included and classified into 2 groups. The propensity scores were used to balance covariates in baseline characteristics including age, body weight, SCr and CrCl. The Mann-Whitney U test was used to compare differences of apixaban level and anti-FXa activity between the control and apixaban with amiodarone group. Bleeding outcomes were expressed as percentages while Chi-square tests were used to identify statistical differences. Two-tailed p-value<0.05 was considered statistically significant.

## Results

### Patient characteristics

During the study period, 109 NVAF patients were identified from the electronic database of the hospital. All patients were already receiving apixaban with or without amiodarone before enrollment. A total of 41 (37.61%) of the patients were excluded because the patients denied to sign informed consent, because they denied venipuncture in 27 patients, CrCl less than 15 mL/min or required hemodialysis in 12 patients, and inability to conduct self-care in 2 patients. Therefore, a total of 68 patients were enrolled. There were 57 and 11 patients receiving apixaban monotherapy and apixaban with amiodarone groups, respectively. The propensity score matching was used to balance covariates of patients in apixaban monotherapy group that there were 22 patients in this group. This study classified the patients into two groups: 11 patients were treated with apixaban with amiodarone, and 22 patients were treated with apixaban monotherapy (control group). Fig 1 shows the details of the patient flow.

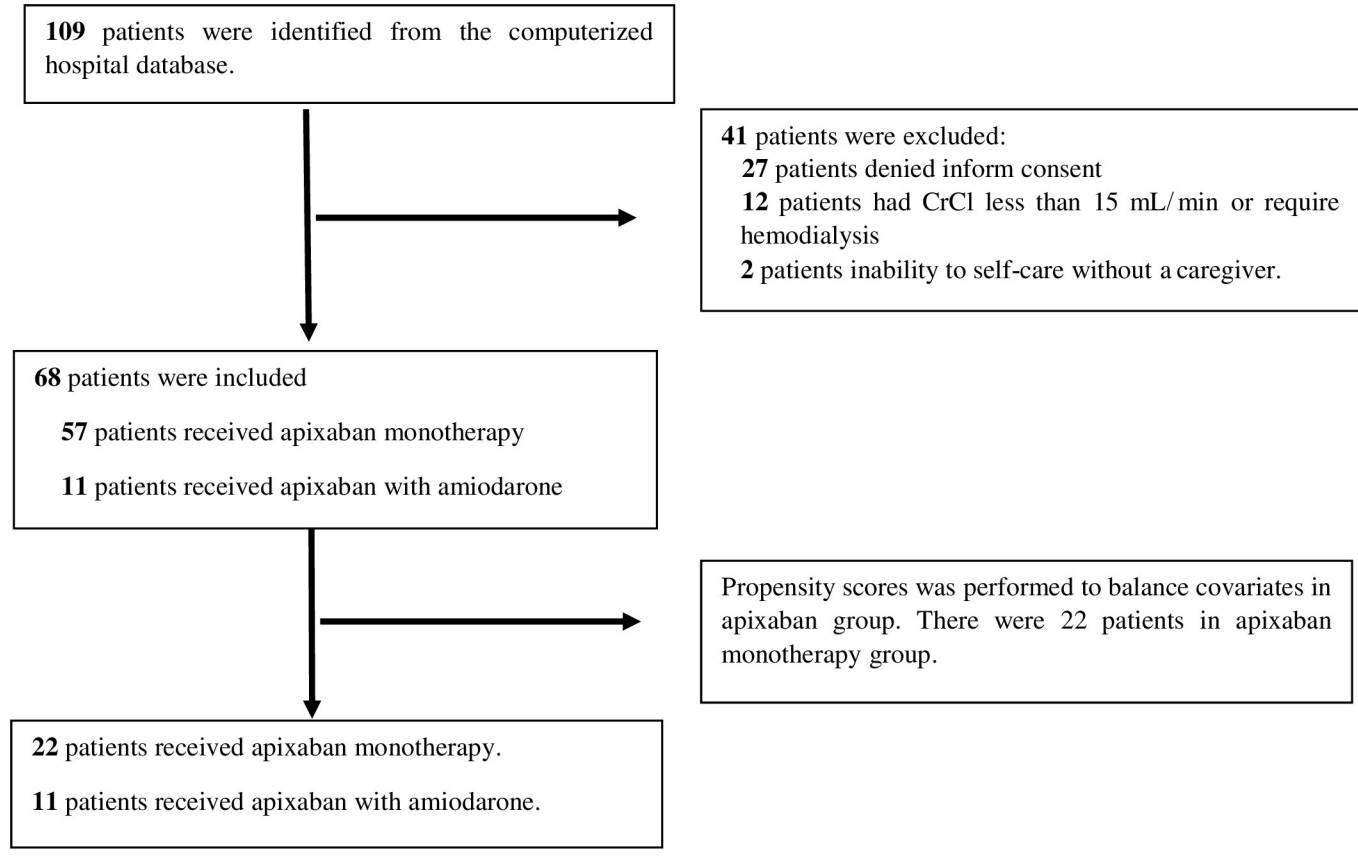

**Fig 1. The details of the patient flow.**

Table 1 provides the demographic and clinical characteristics of 33 patients included in the study. Their mean age was 71.82 ± 10.96 years, 27% of the patients were >80 years old, and 63.64% were male. Patients with body weight ≤ 60 kg accounted for 27.27%. Hypertension (87.88%), dyslipidemia (70.00%), and heart failure (48.48%) were the three most common comorbidities. The treatment duration of apixaban was 9 (IQR 15,39) months. The CrCl was 57.50 ± 23.88 mL/min. The median of CHA$_2$DS$_2$-VASC and HAS-BLED scores were 5 and 2 points, respectively. A total of 69% of the patients (23 patients) received the standard dose of apixaban (15 and 8 patients in the control and apixaban with amiodarone groups, respectively. Overall, most baseline characteristics were similar between the groups except for the level of alanine transaminase (ALT), which was significantly higher in the apixaban with amiodarone group. The median amiodarone dose was 200.0 (IQR: 150–312) mg/day, and the duration of use was 4.3 (2.0–6.9) months.

The median times of the apixaban peak and trough were similar between the groups. Overall, the median times of apixaban were 3 (IQR: 2.83–3.05) and 12.00 (IQR: 11.83–12.50) hours for peak and trough levels, respectively. All patients in the median [5th–95th percentile] were measured for apixaban plasma levels. The C$_{trough}$ reached 108.49 [78.10–171.52] and 162.05 [87.94–292.88] µg/L in the control and apixaban with amiodarone groups, respectively (p = 0.028). In addition, the C$_{peak}$ values were 175.36 [122.94–332.34] and 191 [116.88–488.21] µg/L in the control and apixaban with amiodarone groups, respectively (p = 0.375).

Results of apixaban plasma level and anti-FXa activity in control and apixaban with amiodarone groups are shown in Table 2. The anti-FXa activity between the control and apixaban

**Table 1. Demographic and clinical characteristics of 33 patients included in the study.**

| Parameters | Overall | Control (n = 22) | Apixaban with amiodarone (n = 11) | P-value |
|---|---|---|---|---|
| Age (years) | 71.82 ± 10.96 | 71.95 ± 10.57 | 71.55 ± 12.22 | 0.962 |
| Male sex | 21(63.64) | 14(63.64) | 7(63.64) | 1.000 |
| Body weight (kg) | 68.38 ± 15.49 | 69.79 ± 14.95 | 65.21 ± 16.79 | 0.436 |
| SCr (mg/dL) | 1.09 [0.92–1.35] | 1.12 [0.92–1.37] | 1.02 [0.99–1.30] | 0.920 |
| CrCl (mL/min) | 57.50 ± 23.88 | 57.76 ± 22.42 | 56.99 ± 27.72 | 0.938 |
| AST (U/L) | 24.20 [21.00–32.10] | 23.70 [19.70–31.45] | 30.70 [21.65–55.70] | 0.244 |
| ALT (U/L) | 19.80 [15.00–30.80] | 18.30 [14.48–24.38] | 30.8 [20.40–38.85] | 0.023[†] |
| $CHA_2DS_2VASC$ score | 5[2–6] | 5[2–6] | 5[3–6] | 0.938 |
| HAS-BLED score | 2[1–2] | 2[1–2] | 1[1–3] | 0.953 |
| Prior bleeding | 11(33.33) | 8(36.27) | 3(27.27) | 0.709 |
| Apixaban dose (mg twice daily) | 5[2.5–5] | 5[2.5–5] | 5[3.75–5] | 0.811 |
| Underlying diseases | | | | |
| Hypertension | 29(87.88) | 20(90.91) | 9(81.82) | 0.586 |
| Dyslipidemia | 21(70.00) | 14(63.64) | 7(63.64) | 0.703 |
| Heart failure | 16(48.48) | 10(45.45) | 6(54.55) | 0.622 |
| Type II DM | 15(45.45) | 10(45.45) | 5(45.55) | 1.000 |
| Anemia | 14(42.42) | 8(36.36) | 6(54.55) | 0.459 |
| CKD | 10(30.30) | 6(27.27) | 4(36.36) | 0.696 |
| Clopidogrel | 3(9.09) | 2(18.18) | 1(4.55) | 0.252 |

with amiodarone groups, including the peak and trough, was not statistically significant. The trough anti-FXa activities were 1.24 [0.80–2.00] IU/mL in the control group and 1.94 [0.83–2.74] IU/mL in the apixaban with amiodarone group (p = 0.082). The anti-FXa activities at the peak were 1.96 [1.60–2.95] and 2.42 [1.32–4.30] IU/mL in the control and apixaban with amiodarone groups, respectively (p = 0.369).

The apixaban plasma levels were approximately 13.60% and 18.20% out of the expected range in the control and apixaban-with-amiodarone groups, respectively. We analyzed the association between apixaban plasma levels above the expected ranges with or without amiodarone. The results showed that amiodarone may increase the $C_{trough}$ and $C_{peak}$ of apixaban.

**Table 2. Apixaban plasma level and anti-FXa activity in control and apixaban with amiodarone groups.**

| Parameters | Overall (N = 33) | Control (n = 22) | Apixaban with amiodarone (n = 11) | P-value |
|---|---|---|---|---|
| Time to peak (hours) | 3.00 [2.83 3.05] | 3.00 [2.96–3.08] | 3.00 [2.63–3.00] | 0.207 |
| Last dose to trough (hours) | 12.00 [11.83–12.50] | 12.04 [12.00–12.50] | 11.50 [11.46–12.70] | 0.170 |
| $C_{trough}$ (µg/L) | 120 [77.56–216.92] | 108.49 [78.10–171.52] | 162.05 [87.94–292.88] | 0.028[†] |
| $C_{peak}$ (µg/L) | 183.05 [109.81–378.85] | 175.36 [122.94–332.34] | 191 [116.88–488.21] | 0.375 |
| $Xa_{trough}$ (IU/mL) | 1.32 [0.79–2.32] | 1.24 [0.80–2.00] | 1.94 [0.83–2.74] | 0.082 |
| $Xa_{peak}$ (IU/mL) | 2.03 [1.39–3.11] | 1.96 [1.60–2.95] | 2.42 [1.32–4.30] | 0.369 |

However, the co-administration of apixaban with amiodarone exhibited no significant association with the $C_{trough}$ [odds ratio (OR): 2.05 (95% confidence interval (CI): 0.02–172.29), p = 1.000] and $C_{peak}$ [OR: 2.16 (95% CI: 0.14–34.40), p = 0.586] above the expected range, as shown in Table 3.

The median of follow-up periods of bleeding outcomes were 12 (IQR:12.00–12.00) months, 7 patients (21.21%) of patients presented bleeding. Major bleeding was observed in 2 patients (6.06%), and CRNMB bleeding was found in 5 patients (15.15%). In certain cases, major bleeding involved the gastrointestinal tract and brain. Among the total of 7 patients in the bleeding group; 2 patients received apixaban with amiodarone while 5 patients received apixaban monotherapy.

In those patients with bleeding, $C_{peak}$ was 148.69 [141.92–283.45] µg/L and $C_{trough}$ was 111.48 [102.67–159.86] µg/L. Table 4 summarizes the apixaban plasma levels classified by bleeding events.

## Discussion

In real-life practice, amiodarone is usually used with apixaban for rate control in AF patients. Based on its pharmacokinetics data, amiodarone is a substrate of CYP3A4 and CYP2C8. In the same manner, amiodarone is an inhibitor of CYP3A4 and P-gp. CYP3A4 is an important metabolizer for apixaban; approximately 20–25% of apixaban is used as a substrate for P-gp [12]. Therefore, the concomitance between amiodarone and apixaban may affect the apixaban plasma level and increase the risk of bleeding. The principal findings of this study indicated the peak and trough of apixaban plasma level in the apixaban-with-amiodarone group was higher than that in the apixaban-monotherapy group and statistically significant in $C_{trough}$. In addition, the concomitance of apixaban-with-amiodarone was not related to the above-expected-level results. This study reported the first real-world evidence investigating the apixaban plasma levels in patients who received the combination of apixaban and amiodarone in the Southeast Asia region.

The data reported from VigiBase, the World Health Organization database of spontaneous safety reports, showed that amiodarone has no significant effect on the AUC of apixaban [17]. Meanwhile, a previous observational study revealed that the apixaban trough concentration upon co-administration of amiodarone was elevated compared with apixaban alone [15]. In addition, Gulila M., et al. showed that impaired-function variant ABCG2 c.421C > A, sex, and co-administration with amiodarone, which moderates the CYP3A4 and P-gp inhibitors, can be predictive of a high apixaban level [18]. These results were consistent with those of Cirincione B., et al.'s study [21]. However, real-world research reported that apixaban, concomitant with CYP3A or P-gp inhibitor drugs, was not related to the above expected levels of apixaban in patients with AF [23]. Many studies have shown the increased apixaban plasma levels caused by amiodarone; while the doses of amiodarone used in this study were less than those applied in Gulilat M, et al.'s study, in which the average doses of amiodarone were 200 and 400 mg/day [18]. However, data about the effect of amiodarone on other pharmacokinetic profiles of apixaban are limited. Moreover, many factors can affect the apixaban plasma level such as age, body weight, renal function and potential drug interaction (especially azole anti-fungals) [24]. Therefore, these factors have been recommended for apixaban dose adjustment in NVAF patients as follows: age ≥ 80 years, body weight ≤ 60 kg and SCr ≥ 1.5 mg/dL. Patients meeting two of the above-mentioned criteria received a reduced dose of apixaban (2.5 mg twice daily) [14]. A previous study reported that apixaban exposure increased by 30% in the low-body-weight group and decreased by 20% in the high-body-weight group when compared with a reference weight group [25]. However, the magnitude of these changes was not

**Table 3. Association between apixaban plasma levels above the expected ranges with or without amiodarone.**

| Apixaban level above expected ranges | Apixaban-with-amiodarone (n = 11) | Control (n = 22) | OR (95%CI) | P-value |
|---|---|---|---|---|
| $C_{trough}$, n(%) | 1(9.09) | 1(4.55) | 2.05 (0.02–172.29) | 1.000 |
| $C_{peak}$, n(%) | 2(18.18) | 2(9.09) | 2.16 (0.14–34.40) | 0.586 |

considered clinically meaningful, and no dose adjustment based on body weight alone has been recommended. Approximately 27% of apixaban is excreted by the renal system [7]. In addition, many studies have reported that renal insufficiency can be increased by apixaban plasma level and AUC of apixaban [21, 26]. The baseline characteristics of patients from another study were similar to those in this study (age, sex, body mass index, and CrCl) [23]. This study showed the statistical significance of ALT between apixaban with amiodarone and apixaban monotherapy. Likewise, apixaban exposure was not significantly modified by mild and moderate hepatic impairment (Child–Pugh A and B), but apixaban was contraindicated in Child–Pugh C.

Currently, the bleeding outcomes of the interaction of apixaban with amiodarone are limited. Flaker G., et al. analyzed the influence of amiodarone on the outcomes of the ARIS-TOTLE trial, which compared warfarin and apixaban for the prevention of systemic embolism and stroke among NVAF patients [27]. These results showed no significant differences in the incidences of bleeding events for apixaban with or without amiodarone, and the major bleeding rates were 1.86%/year and 2.18%/year for apixaban-with-amiodarone and apixaban-monotherapy groups, respectively. The median plasma levels in each group differed by approximately 15%, which caused difficulty in identifying a therapeutic range [27]. However, no head-to-head comparison has been conducted for each anticoagulant with or without amiodarone [26]. The median plasma levels in each group differed by approximately 15%, which caused difficulty in identifying a therapeutic range [26]. In addition, some case reports showed the hemopericardium drug–drug interaction between apixaban and amiodarone, and the patient's advanced age and borderline creatinine were possible risk factors [28]. Similarly, a retrospective cohort study of the elderly reported no increased bleeding risk in novel oral anticoagulants (NOACs) concomitant with amiodarone [29]. Those results were consistent with our analysis, which showed that receiving apixaban in combination with amiodarone was not related to a bleeding event in one year. However, this finding is controversial with Chang SH, et al.'s, study that reported concurrent use of amiodarone with NOACs showed a significant increase in adjusted incidence rates of major bleeding than NOACs alone (38.09 per 1000 person-years for NOACs use alone and 52.04 per 1000 person-years for NOACs with amiodarone (difference, 13.94 [99%CI, 9.76–18.13]). Unfortunately, the data extracted from that study were not classified into each NOACs [19]. Likewise, retrospective cohort studies reported that rivaroxaban and apixaban concomitant with amiodarone increased the bleeding risk [30].

**Table 4. Apixaban plasma levels classified by bleeding events.**

| Apixaban level (μg/L) | Non-bleeding event (n = 26) | Bleeding events (n = 7) | P-value |
|---|---|---|---|
| $C_{trough}$ | 121.33 [98.78–162.05] | 111.48 [102.67–159.86] | 0.843 |
| $C_{peak}$ | 187.74 [162.92–239.80] | 148.69 [141.92–283.45] | 0.449 |

However, no head-to-head comparison has been conducted for each anticoagulant with or without amiodarone. The results in this study, showed that bleeding occurred in apixaban monotherapy group more than apixaban with amiodarone group which might be due to the differences of ALT in the baseline characteristics between the two groups. The apixaban monotherapy group had higher ALT than the apixaban amiodarone group.

A real-world pilot prospective study reported significantly higher apixaban plasma levels in the bleeding group compared with the non-bleeding group [31]. This study's finding was not consistent with that finding, and showed $C_{peak}$ and $C_{trough}$ of apixaban plasma level in the non-bleeding group was higher than the bleeding group.

Several limitations need to be addressed. First, this study used a small sample size, and did not evaluate the relationship between bleeding outcomes and apixaban plasma levels, including the anti-FXa activity, in the apixaban with amiodarone group. Therefore, multicenter or larger sample sizes should be used in future studies. Second, intra-individual variability in apixaban plasma levels was reported in a previous study. Thus, a randomized control trial should be conducted. Third, some bleeding outcomes were not recorded in the hospital database, and the short duration of follow-up in this study may lead to an underestimation. This study estimated the $C_{peak}$ for 2–4 hours. No pharmacokinetic data were available at that time to determine the actual peak for each patient. Finally, due to apixaban plasma levels were not measured in bleeding events later. Therefore, as a result apixaban plasma levels in the bleeding group cannot definitively be confirmed that they were over the expected range.

## Conclusion

Current knowledge from this study showed, that amiodarone affected apixaban plasma at the peak and trough levels, especially trough levels were statistically significantly increased. However, this study could not summarize that amiodarone with apixabn was associated with increased bleeding outcomes, compared to apixaban monotherapy users. However, close monitoring of bleeding symptoms in patients who use concomitant apixaban with amiodarone is necessary.

## Acknowledgments

The authors would like to thank all the staff of the Pharmacy Department and all cardiologists of the Division of Cardiology, Department of Internal Medicine, Phramongkutklao Hospital and Hematology Laboratory, Division of Hematology, Department of Internal Medicine, Phramongkutklao Hospital. The authors thank Stephen Pinder a native-speaking medical English specialist for comprehensive language review of our manuscript to an international level.

## Author Contributions

**Conceptualization:** Sutee Limcharoen, Sarawuth Limprasert, Pornwalai Boonmuang.

**Formal analysis:** Sutee Limcharoen, Pornwalai Boonmuang.

**Funding acquisition:** Pornwalai Boonmuang.

**Investigation:** Sutee Limcharoen, Pornwalai Boonmuang, Weerayuth Saelim.

**Methodology:** Sutee Limcharoen, Pornwalai Boonmuang, Manat Pongchaidecha.

**Project administration:** Sutee Limcharoen, Pornwalai Boonmuang.

**Supervision:** Pornwalai Boonmuang.

**Validation:** Sutee Limcharoen, Sarawuth Limprasert, Pornwalai Boonmuang, Manat Pongchaidecha, Juthathip Suphanklang, Weerayuth Saelim, Wichai Santimaleeworagun, Piyarat Pimsi.

**Visualization:** Pornwalai Boonmuang.

**Writing – original draft:** Sutee Limcharoen, Sarawuth Limprasert, Pornwalai Boonmuang.

**Writing – review & editing:** Sutee Limcharoen, Pornwalai Boonmuang.

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
