## [Decision Letter · Decision Letter 0]

21 Jun 2023

PONE-D-23-05377Does amiodarone impact on apixaban levels? A significant of trough concentration apixaban among patients with non-valvular atrial fibrillationPLOS ONE

Dear Dr. boonmuang,

Thank you for submitting your manuscript to PLOS ONE. After careful consideration, we feel that it has merit but does not fully meet PLOS ONE’s publication criteria as it currently stands. Therefore, we invite you to submit a revised version of the manuscript that addresses the points raised during the review process.

We look forward to receiving your revised manuscript.

Kind regards,

Nienke van Rein

Academic Editor

PLOS ONE

Journal Requirements:

Reviewers' comments:

Reviewer's Responses to Questions

**Comments to the Author**

1. Is the manuscript technically sound, and do the data support the conclusions?

Reviewer #1: Partly

Reviewer #2: Partly

2. Has the statistical analysis been performed appropriately and rigorously? 

Reviewer #1: No

Reviewer #2: Yes

3. Have the authors made all data underlying the findings in their manuscript fully available?

Reviewer #1: Yes

Reviewer #2: No

4. Is the manuscript presented in an intelligible fashion and written in standard English?

Reviewer #1: No

Reviewer #2: No

5. Review Comments to the Author

Reviewer #1: Limcharoen et al included 33 NVAF patients who were receiving apixaban and compared the medication consternation and bleeding risk between those who received concurrent amiodarone and those who did not. The found patients with concurrent amiodarone use had significantly higher trough concentration of apixaban, but was not associated with bleeding risk. There are a few issues to address/clarify:

1. The study design was confusing. The section Study participants suggests the patients were already receiving apixaban with or without amiodarone before enrollment since adherence to the use of these medications within 7 days before enrollment were examined (Page 3), and the patients were further retrospectively categorized according to use of amiodarone.

However, in the result section (Page 5), the authors stated “We classified the patients into two groups by propensity score matching (1: 2): 11 patients were treated with apixaban-with- amiodarone, and 22 were subjected to apixaban-monotherapy (control group).” This seems to suggest the assignment of amiodarone was actually conducted by the authors. If this is the case, I wonder whether it is ethical to prescribe amiodarone without a clear (prespecified) indication. More details about this should be provided. Also, for a prospective interventional study, why not consider a random assignment?

2. In the manuscript it was stated that the study was conducted from January 2021; however, according to the TCTR registry (TCTR20210311005), the first enrollment was actually 30 December 2020. The authors should keep the manuscript consistent with the actual enrollment. I also noticed that study was retrospectively registered, which should be mentioned in the manuscript.

3. Details about the propensity score matching should be described. If the two groups were assigned by propensity scores, it would not be surprised to see “most baseline 190 characteristics were similar between the groups”.

4. For the investigated clinical outcomes, how were they identified? How long were the patients followed for? These details should also be clearly described.

5. In the result section, it was mentioned “During the 1-year study period following up…” Since the study was conducted between January 2021 and January 2022, would this mean the majority of the included patients were actually followed for less than one year? If this is the case, censoring should be taken into account for the clinical outcome analysis.

In the registered protocol, the anticipated last enrollment would be June 30 2022, and the completion date of the study would be December 2022. I wonder what made the authors terminate the study earlier.

6. The planned sample size was 130 according to the TCTR registry (TCTR20210311005), while only 33 were included only (from 109 patients). How was the sample size estimated? What caused the change?

7. Relevant studies about potential impact of amiodarone use on bleeding risk among AF patients receiving apixaban were not well discussed. In a recent large-scale study (PMID 37216662), concurrent use of amiodarone was found to be associated with increased risk of bleeding-related hospitalizations. I think this further suggests the current study might be underpowered to observe any difference in bleeding outcomes.

8. Apixaban plasma levels were presented as median (5th-95th percentiles). I wonder why the authors preferred this, instead of median with IQR (i.e., 25-75th percentiles).

9. For analysis of the bleeding outcomes, occurrence of the bleeding outcomes was not directly compared between the two groups (i.e., apixaban without amiodarone versus apixaban with amiodarone). Instead, the authors compared amiodarone use in patients with bleeding events to those without. Such an analysis seems rather rare to see, and I think here the comparison lost the benefit of confounding control by propensity score matching.

10. Many analyses (particularly about the bleeding outcomes) were not mentioned at all in the section Statistical analysis.

11. “Notably, the peak and trough of apixaban plasma level in the apixaban-with-amiodarone group was higher than that in the apixaban-monotherapy group and statistically significant.” This statement was incorrect, as the authors found no significant difference in apixaban Cpeak. This issue applies to the conclusion “Amiodarone affected the peaks and troughs of apixaban plasma levels”.

12. In the introduction, the authors stated “… but no pharmacokinetic data are available for the apixaban group” However, in the discussion, they gave many examples (e.g., reference 19, 15, 20) about existing studies that investigated pharmacokinetic data of apixaban.

13. Minor points:

1) Title: There is a grammatical error in the title “A significant of …”. Did the authors mean “significance of …”?

2) Keywords: Consider to add Amiodarone into the keywords.

3) Ethics Statement: There is a grammatical error “All eligible provided …”

4) Abstract: Avoid using causal wordings in the conclusion.

5) P value was missing in the last sentence of the results in the abstract. There seems also a typo “in the couple”.

6) The reference 4 did not fully support the statement “The three major goals of treatment are …”.

7) Proofreading is needed about the English language and use of punctuations.

Reviewer #2: In the study the authors compared the apixaban levels and bleeding outcomes between apixaban monotherapy and apixaban with amiodaron therapy groups. The research question is interesting and relevant. Furthermore, the prospective design of the study improves the quality. However, I have some concerns and questions. Most importantly, the sample size of only 7 bleeds is too small to draw statistical conclusions on the association between bleeding outcomes and the concomitant use of amiodaron.

General:

- The English language in the paper could be improved. Importantly, the title of the paper should be altered to improve the reading quality.

Furthermore, several sentences are not correct. Please check your full paper on English language. For instance, page 2 line 75-76 could be changed to ‘Theoretically, the pharmacokinetic interaction between apixaban and amiodaron may result in increased apixaban plasma levels and bleeding.’

Abstract:

- Page 1, line 37. please clarify ‘couple’ group.

Introduction:

- Page 2, line 79. I regard plasma peak/trough concentrations also as pharmacokinetic data. So your statement that no pharmacokinetic data are available for the apixaban group is not correct, please change the sentence.

- In literature I found a paper of Chang et al, which described bleeding outcomes of patients using NOACs and amiodaron. This would be an interesting paper to add to the introduction.

Chang SH, Chou IJ, Yeh YH, Chiou MJ, Wen MS, Kuo CT, See LC, Kuo CF. Association Between Use of Non-Vitamin K Oral Anticoagulants With and Without Concurrent Medications and Risk of Major Bleeding in Nonvalvular Atrial Fibrillation. JAMA. 2017 Oct 3;318(13):1250-1259. doi: 10.1001/jama.2017.13883. PMID: 28973247.

In the discussion paragraph the authors cite this and more other papers. Please phrase the introduction differently, so that previous knowledge from literature and the addition/relevancy of your work is more clear.

Methods

- Page 3, line 20-22. I think you mean that the recommended dose is 5 mg twice daily. For patients with at least two of the noted characteristics the dose is 2.5 mg twice daily. Or is the dosing guideline different for Thailand? Then the results are less applicable for other countries?

- How was the anti Xa level measured?

- What was used as definition for minor bleeding?

Results

- page 4 line 73 – page 5 line 77. Please clarify why 62 patients had enrollment denials. Furthermore, not 38% of the patients were excluded, but in total 76 of the 109 patients were excluded (70%). This is a high number, do the authors expect bias?

- A total of 33 patients is a relatively small group to show statistical differences. Was a power analysis done to investigate which number of patients is necessary?

Especially, the univariate and multivariate analysis of the bleeding is done with only 7 bleeds. In my view, it is not valid to draw the conclusion that concomitant use of apixaban amiodaron is not associated with more bleeding based on this small number.

- Table 1, please explain what bleeding history means.

- Please add the median and IQR of apixaban dose and treatment duration.

- Were the patients still using apixaban and amiodaron when bleeding was observed during the 1-year follow up.

- Did the dose of amiodaron or apixaban change during the one year follow up? Now the trough and peak levels measured in the beginning are compared with bleeding events later that year.

- Table 2: trough to peak. The time from apixaban administration to peak describes the time to peak better than the time from trough to peak. Or is always directly after the trough the dose administrated?

Conclusion:

- In the conclusion you state the amiodaron affects the apixaban levels. However, in your results you state that only the trough levels are statistically significantly different. Please extend your conclusions so it concludes that higher peak and trough levels were found when amiodaron was used, but that this was statically significant for trough levels in the small sample size.

- As stated before, the conclusion that no clinically significant bleeding event occurred when comparing the control and the amiodaron group, is too strong for the small sample size.

6. PLOS authors have the option to publish the peer review history of their article (what does this mean?). If published, this will include your full peer review and any attached files.

Reviewer #1: No

Reviewer #2: No

---

## [Author Response · Author response to Decision Letter 0]

16 Aug 2023

Reviewer 1: Thank you for valuable comments and your suggestions. I have incorperated all of your suggestions in to my revision.

Reviewer 2: Thank you for valuable comments and your suggestions. I have incorperated all of your suggestions in to my revision.

---

## [Decision Letter · Decision Letter 1]

4 Oct 2023

PONE-D-23-05377R1Does amiodarone impact on apixaban levels? The effect of amiodarone on apixaban level among Thai patients with non-valvular AFPLOS ONE

Dear Dr. boonmuang,

Thank you for submitting your manuscript to PLOS ONE. After careful consideration, we feel that it has merit but does not fully meet PLOS ONE’s publication criteria as it currently stands. Therefore, we invite you to submit a revised version of the manuscript that addresses the points raised during the review process.

We look forward to receiving your revised manuscript.

Kind regards,

Nienke van Rein

Academic Editor

PLOS ONE

Reviewers' comments:

Reviewer's Responses to Questions

**Comments to the Author**

1. If the authors have adequately addressed your comments raised in a previous round of review and you feel that this manuscript is now acceptable for publication, you may indicate that here to bypass the “Comments to the Author” section, enter your conflict of interest statement in the “Confidential to Editor” section, and submit your "Accept" recommendation.

Reviewer #1: (No Response)

Reviewer #2: (No Response)

2. Is the manuscript technically sound, and do the data support the conclusions?

Reviewer #1: Partly

Reviewer #2: Partly

3. Has the statistical analysis been performed appropriately and rigorously? 

Reviewer #1: Yes

Reviewer #2: No

4. Have the authors made all data underlying the findings in their manuscript fully available?

Reviewer #1: No

Reviewer #2: Yes

5. Is the manuscript presented in an intelligible fashion and written in standard English?

Reviewer #1: Yes

Reviewer #2: No

6. Review Comments to the Author

Reviewer #1: I have the following comments for the revised manuscript:

1. According to the revised version, the study presented in the manuscript was a prospective observational study, in which the study population was first included to draw blood samples for measuring apixaban levels, and then they were followed for bleeding events. This suggested the exposure namely concurrent use of amiodarone was not assigned by the authors, instead, based on the actual use of amiodarone before inclusion (as described in the method section).

According to Figure 1, 33 patients were included for propensity score matching (1:2) by “age, body weight, SCr and CrCl”, and it happened to be 11 patients and 22 patients who were finally matched successfully as the two groups. Also, sex was not mentioned as a covariate for calculating the propensity score, but according to Table 1, the two groups had exactly the same sex distribution. I doubt whether this is indeed possible (given such a small sample size).

Besides, details about how the propensity score was calculated and how the matching was performed were still absent.

2. Minor points:

1) Abstract- Methods: “in patients …” It is better to mention what patients here (i.e., NVAF patients).

2) Abstract- Results: “Likewise, Cpeak showed no difference in the non-bleeding (148. 69 [141. 92–283. 45] μg/L) and bleeding (187.74 [1162.92–238.80] μg/L) groups, p=449.” The results are not consistent with Table 4, and there are also several typos, such as “1162.92”, “p=449”.

3) Abstract- Conclusions: The conclusion is with causal wordings: “… affects…”, “… causes…” With the used study design, the current study could not draw such conclusions.

4) Methods: The statistical analysis section still did not cover all the analyses the authors presented in the results.

5) Results: Median follow-up should be provided in the result section.

6) Result: “The results showed that the Ctrough and Cpeak were increased by amiodarone.” Since the difference (Table 3) was not statistically significant, I think it is not appropriate to make such a statement.

Reviewer #2: Thank you for submitting a new version of the manuscript and the adjustments that are made. Nonetheless, some concerns that were raised in the previous reviewer comments remain and are not (fully) adjusted in this version.

1. You added statements about the small sample size in the discussion. However, in my opinion your sample size it still too small to perform statistical test on the relation between bleeding and the concomitant use of amiodaron with apixaban. You can only use descriptive statistics for the bleeding events that occurred. This makes it also not possible to state there was no significant difference in bleeding between the two groups.

2. Page 3, line 95. Thank you for adding more relevant literature to the introduction and changing the statement that no pharmacokinetic data was available. Though, while reading this adjusted introduction you get the impression that most studies did not show an increase in risk of stroke and bleeding. While for instance the following two large observational trials showed an increased risk of bleeding when using amiodaron and apixaban. (10.1001/jama.2017.13883, 10.7326/M22-3238)

3. You added the apixaban treatment duration of median 9 months. Were patients censored when the stopped using apixaban? Is a bleeding event still relevant for your research question if it occurred after treatment with apixaban?

4. The English language in the manuscript is improved. However, the manuscript still includes some grammatical errors.

Other comments, based on the corrections made after input of reviewer 1.

1. During inclusion of this prospective observational study 33 NVAF patient were enrolled that used apixaban with or without amiodaron. It is correct that from the patients that fulfilled the inclusion criteria (33 patients), exactly 2/3 used apixaban monotherapy and 1/3 used apixaban with amiodaron? This seems like an almost too perfect coincidence?

2. The authors explain to reviewer 1 that censoring of data was conducted when the bleeding outcome occurred or at 1 year after the blood samples were collected. Bleeding is an event that can occur multiple times, would it not be more appropriate to follow all patients 1 year and take repeated events into account?

Minor:

1. #page 4, line 45-47. The SmPC of apixaban states that for NVAF the recommended dose of is 5 mg twice daily. When patients have at least 2 of the following characteristics ( age ≥ 80 years, body weight ≤ 60 kg, or serum creatinine (SCr) ≥ 1.5 mg/dL) the dose is 2.5 twice daily. I wonder why you stated exactly the opposite? In the discussion (page 11, line 79) you state the dose correctly.

https://www.ema.europa.eu/en/documents/product-information/eliquis-epar-product-information_en.pdf

7. PLOS authors have the option to publish the peer review history of their article (what does this mean?). If published, this will include your full peer review and any attached files.

Reviewer #1: No

Reviewer #2: No

---

## [Author Response · Author response to Decision Letter 1]

19 Oct 2023

Reviewer 1: I have incorperated all of your suggestions in to my revision. They were very help.Thank you very much for your help.

Reviewer 1: I have incorperated all of your suggestions in to my revision. They were very help.Thank you very much for your help.

---

## [Editor Report · Decision Letter 2]

7 Nov 2023

PONE-D-23-05377R2Does amiodarone impact on apixaban levels? The effect of amiodarone on apixaban level among Thai patients with non-valvular AFPLOS ONE

Dear Dr. boonmuang,

Thank you for submitting your manuscript to PLOS ONE. After careful consideration, we feel that it has merit but does not fully meet PLOS ONE’s publication criteria as it currently stands. Therefore, we invite you to submit a revised version of the manuscript that addresses the points raised during the review process.

Some comments of the reviewers were not properly addressed. Could you please reconsider these:

- 'You added statements about the small sample size in the discussion. However, in my opinion your sample size it still too small to perform statistical test on the relation between bleeding and the concomitant use of amiodarone with apixaban. You can only use descriptive statistics for the bleeding events that occurred. This makes it also not possible to state there was no significant difference in bleeding between the two groups.'

The abstract still contains statements about the bleedings. This statement should be removed. Furthermore, please check the manuscript for statements regarding bleeding (7 bleedings and 2 major bleedings are not high enough numbers to find a significant association or say anything about effects of amiodaron with apixaban versus apixaban monotherapy).

- The answer 'Thank you for your suggestion. We added this sentence “However, two large observational trials showed an increased risk of bleeding when using amiodarone with apixaban. ” in introduction section, page 3 (lines 85-86).'

Observational studies cannot be trials (i.e., if they were trials, they were not observational), please rephrase.

We look forward to receiving your revised manuscript.

Kind regards,

Nienke van Rein

Academic Editor

PLOS ONE

---

## [Author Response · Author response to Decision Letter 2]

22 Nov 2023

Reviewer 1: I have incorperated all of your suggestion into my revision. They were very helpful.

Reviewer 2: I have incorperated all of your suggestion into my revision. Thank you for your valuable comments.

---

## [Editor Report · Decision Letter 3]

24 Nov 2023

Does amiodarone impact on apixaban levels? The effect of amiodarone on apixaban level among Thai patients with non-valvular AF

PONE-D-23-05377R3

Dear Dr. boonmuang,

We’re pleased to inform you that your manuscript has been judged scientifically suitable for publication and will be formally accepted for publication once it meets all outstanding technical requirements.

Kind regards,

Nienke van Rein

Academic Editor

PLOS ONE
---

## [Editor Report · Acceptance letter]

8 Jan 2024

PONE-D-23-05377R3 

PLOS ONE

Dear Dr. boonmuang, 

I'm pleased to inform you that your manuscript has been deemed suitable for publication in PLOS ONE. Congratulations! Your manuscript is now being handed over to our production team.

Kind regards, 

on behalf of

Dr. Nienke van Rein 

Academic Editor

PLOS ONE